# Advancing Field-Based Vertical Jump Analysis: Markerless Pose Estimation vs. Force Plates

**DOI:** 10.3390/life14121641

**Published:** 2024-12-11

**Authors:** Jelena Aleksic, David Mesaroš, Dmitry Kanevsky, Olivera M. Knežević, Dimitrije Cabarkapa, Lucija Faj, Dragan M. Mirkov

**Affiliations:** 1Faculty of Sport and Physical Education, University of Belgrade, 11000 Belgrade, Serbia; jelena.aleksic@fsfv.bg.ac.rs (J.A.); olivera.knezevic@fsfv.bg.ac.rs (O.M.K.); 2School of Electrical Engineering, University of Belgrade, 11000 Belgrade, Serbia; davidmesaros9905@gmail.com; 3New Stream, Kfar Sava 4465141, Israel; dima@annoviz.com; 4Jayhawk Athletic Performance Laboratory–Wu Tsai Human Performance Alliance, Department of Health, Sport and Exercise Sciences, University of Kansas, Lawrence, KS 66045, USA; dcabarkapa@ku.edu; 5Faculty of Kinesiology, University of Osijek, 31000 Osijek, Croatia; lfaj250@gmail.com

**Keywords:** vertical jump, AI, MoCap, jump performance, sports diagnostics, performance analysis

## Abstract

The countermovement vertical jump (CMJ) is widely used in sports science and rehabilitation to assess lower body power. In controlled laboratory environments, a complex analysis of CMJ performance is usually carried out using motion capture or force plate systems, providing detailed insights into athlete’s movement mechanics. While these systems are highly accurate, they are often costly or limited to laboratory settings, making them impractical for widespread or field use. This study aimed to evaluate the accuracy of MMPose, a markerless 2D pose estimation framework, for CMJ analysis by comparing it with force plates. Twelve healthy participants performed five CMJs, with each jump trial simultaneously recorded using force plates and a smartphone camera. Vertical velocity profiles and key temporal variables, including jump phase durations, maximum jump height, vertical velocity, and take-off velocity, were analyzed and compared between the two systems. The statistical methods included a Bland–Altman analysis, correlation coefficients (r), and effect sizes, with consistency and systematic differences assessed using intraclass correlation coefficients (ICC) and paired samples *t*-tests. The results showed strong agreement (r = 0.992) between the markerless system and force plates, validating MMPose for CMJ analysis. The temporal variables also demonstrated high reliability (ICC > 0.9), with minimal systematic differences and negligible effect sizes for most variables. These findings suggest that the MMPose-based markerless system is a cost-effective and practical alternative for analyzing CMJ performance, particularly in field settings where force plates may be less accessible. This system holds potential for broader applications in sports performance and rehabilitation, enabling more scalable, data-driven movement assessments.

## 1. Introduction

The countermovement vertical jump (CMJ) is a critical assessment tool for evaluating lower body power, commonly used in sports science and rehabilitation. In controlled laboratory environments, a complex analysis of CMJ performance is possible through marker-based motion capture (MoCap) systems or force plates, providing highly detailed insights into an athlete’s movement mechanics [1,2]. While these technologies are highly accurate, they are also costly and require specialized equipment, making them impractical for widespread or field use [3]. In non-laboratory settings, more traditional equipment such as contact mats, like the Just Jump® mat, and Vertec^®^ systems are commonly used. These systems primarily focus on measuring jump height, which is a key metric in CMJ assessments. For example, the Just Jump^®^ mat has shown a high correlation with lab-based motion capture systems and force plates, providing a reliable and practical alternative for on-the-field testing [4]. Though these tools are more affordable and easier to use in the field, they are limited in the range of metrics they can assess, mainly focusing on jump height without capturing the complex data offered by lab-based systems.

Recent advancements in artificial intelligence have revolutionized human movement science by offering cost-effective and easily accessible markerless pose-estimation alternatives to traditional marker-based systems [5,6,7]. Unlike 3D marker-based setups, which require expensive equipment and complex calibration, 2D markerless solutions can utilize readily available devices such as smartphones or low-cost cameras, making them more practical for field settings in sports and physical therapy [8]. While markerless systems may still face challenges in terms of accuracy compared to their marker-based counterparts, especially for complex multi-planar movements [9,10], they provide significant advantages in terms of simplicity, portability, and cost [11]. Consequently, markerless motion capture technologies are becoming increasingly integrated into both research and applied practice, offering scalable solutions for analyzing human movement across a range of settings [12,13].

These systems have shown promising results in assessing key movement parameters such as joint angles, posture, gait, and overall biomechanics [14,15]. For instance, studies have demonstrated their utility in tracking lower limb kinematics, providing reliable data for both performance enhancement and injury prevention [5,6]. Moreover, Rojas-Lertxundi et al. (2015) reviewed the various applications of 2D motion capture in sports, noting that the reduction in cost and increased accessibility have made these systems viable for athletes and amateur users outside of controlled lab environments [16]. In clinical and rehabilitation settings, markerless systems have been applied to monitor movement patterns in patients recovering from musculoskeletal injuries, offering a non-invasive and practical means for clinicians to assess progress and adjust treatment [17].

Markerless motion capture technologies have also gained significant traction in the analysis of vertical jumps, especially in sports performance and rehabilitation. For example, markerless solutions like the My Jump Lab^®^ [18] offer a simpler, more accessible alternative for assessing key performance metrics in vertical jumps (i.e., jump height) by allowing users to capture video of the jump using only a smartphone camera. Other markerless systems, such as the MMPose-based framework, have been validated against gold-standard 3D motion capture systems, demonstrating excellent agreement in capturing essential variables like angle kinematics, jump height, and eccentric and concentric phase durations (Aleksic et al., 2024) [14]. Studies like those by Aderinola et al. (2023) and Nakano et al. (2020) further emphasize the accuracy and reliability of these technologies for vertical jump analysis, with Aderinola’s smartphone-based markerless system showing high correlations with force plates in jump height measurement [6,19]. As these tools become more refined, they offer a cost-effective solution for coaches, athletes, and therapists to assess and track performance in real-world settings, enhancing both training and rehabilitation outcomes. 

However, despite the growing popularity and accessibility of markerless motion capture technologies for vertical jump analysis, most of these solutions tend to focus only on a few general metrics, such as jump height and ground contact time [18,20]. While these metrics provide valuable insights into overall performance, they represent only a fraction of the kinematic and kinetic data available for a comprehensive understanding of movement. For instance, popular tools like My Jump® app are excellent at quickly and accurately estimating jump height using smartphone video, but they do not delve into more complex variables such as joint angles, velocity profiles, or force production dynamics [8,18]. Similarly, studies like Aderinola et al. (2023) have shown that smartphone-based markerless systems can reliably quantify jump height, but they are limited in their capacity to capture detailed movement phases or biomechanical subtleties that are often necessary for deeper performance assessments and injury prevention strategies [19]. While convenient for quick evaluations, this narrow focus on basic metrics can miss important trends in movement efficiency or compensatory patterns that are critical in both sports training and rehabilitation contexts [21]. As a result, while markerless technologies like My Jump^®^ app are increasingly valuable for providing easily accessible performance data, they are primarily used for more general assessments and may require further development or supplementary tools to capture a wider range of metrics needed for comprehensive movement analysis.

In a recently published study, a more comprehensive markerless solution for analyzing the CMJ was proposed, capable of capturing a set of key performance indicators such as eccentric and concentric phase duration, the lowest CoM point during push-off, maximum vertical velocity, and take-off velocity [14]. While this solution was validated against the gold-standard 3D motion capture system, with excellent inter-device agreement observed, 3D systems remain costly, complex, and impractical for widespread use. Therefore, comparing our markerless system with force plates holds greater practical value, as force plates are more accessible and widely used in real-world sports and rehabilitation settings. This comparison offers an essential step toward making advanced performance analysis more feasible and scalable for broader applications.

The present study was designed to further evaluate the practicality of an innovative markerless solution by comparing it with force plates, a more accessible and widely used tool in sports and rehabilitation settings. The primary objective of this study was to assess the level of agreement between the markerless system and the force plate in measuring key performance indicators of the CMJ. It was hypothesized that the markerless MMPose-based system will demonstrate high agreement with the force plate in calculating key temporal and kinematic variables such as vertical velocity, jump height, and take-off velocity during the CMJ. Such validation is essential for advancing the practical application of markerless motion capture technology in sports performance analysis and rehabilitation, where force plates are frequently utilized.

## 2. Materials and Methods

### 2.1. Experimental Session

This study comprised twelve healthy, physically active (i.e., recreational athletes) participants ($\bar{X}$ ± SD: age = 25.6 ± 3.4 years; height = 179.1 ± 8.3 cm; body mass = 76.0 ± 16.0 kg). More specifically, this sample included 7 males ($\bar{X}$ ± SD: age = 25.3 ± 3.5 years; height = 183.6 ± 4.7 cm; body mass = 80.8 ± 9.4 kg) and 5 females ($\bar{X}$ ± SD: age = 26.4 ± 3.7 years; height = 171.7 ± 7.6 cm; body mass = 68.2 ± 3.6 kg). The inclusion criteria required participants to be physically active (i.e., recreational athletes), engaging in at least 150 minutes of moderate to vigorous exercise per week, with no history of lower limb injuries within the last six months. Additionally, participants needed to have prior experience performing the CMJ to ensure familiarity with the task and minimize variability in technique. All participants provided written informed consent following the guidelines of the University’s Institutional Ethical Review Board (approval #02-848/23-2; date: 5 May 2023) and the Declaration of Helsinki. Based on G*Power calculation (ver. 3.1.9.7; Heinrich-Heine-Universität Düsseldorf, Düsseldorf, Germany), the required sample size for this study was 9 subjects (power = 0.8, significance level = 0.05, effect size = 0.8).

### 2.2. Experimental Protocol

Before the beginning of the testing procedure, participants’ body height and mass were measured using a standard anthropometer (GPM Instruments GmbH, Leuk, Switzerland) and body composition scale (Tanita RD-800-BK, TANITA Medical, Tokyo, Japan). Following this, they completed a standardized warm-up protocol (3 min of light cycling and dynamic stretching exercises) and 2–3 submaximal countermovement jumps for familiarization. Each participant then performed five CMJ trials on dual fixed three-dimensional force platforms (AMTI BP600400, Watertown, MA, USA), which were sampled at 1000 Hz. Participants were instructed to remain still for 2 s before and after the jump and to keep their hands on their hips throughout the entire duration of the jump to reduce upper body influence and ensure consistent lower body movement.

The experimental set-up is illustrated in Figure 1. Each jump was recorded simultaneously using an iPhone 13 (Apple Inc., Cupertino, CA, USA) camera positioned perpendicularly to the floor and capturing the left sagittal plane of the participants. The camera was placed 2 m away from the participants, ensuring that the full body was visible during all jump phases. The tripod height was set at 1.10 meters, approximately at the level of the participant’s abdomen. Videos were recorded at 240 fps (1080p HD). 

To ensure better pose estimation accuracy, participants were asked to wear minimal, form-fitting, dark-colored clothing to create a better contrast with the background [22]. Synchronization of the recording was carried out using a wireless camera remote shutter, initiating the video and force plate recordings simultaneously. All sessions were conducted under controlled laboratory conditions for every participant.

### 2.3. Data Processing

#### 2.3.1. Force Plate Data

The vertical ground reaction (vGRF) force data collected from the force plates was processed using custom MATLAB software ver. 24b (MATLAB and Statistics Toolbox Release 2024b, The MathWorks, Inc., Natick, MA, USA). Participants’ body weight was recorded during the 2-second period when they stood motionless on the force plates before initiating the jump. The beginning of the downward movement (i.e., unweighing phase) was pinpointed as the moment when vGRF dropped below a set threshold of 5 times the standard deviation (SD) of the previously recorded body weight [23]. The braking phase was determined as the phase between the peak negative CoM velocity and the moment the velocity returned to zero, marking the deepest point of the countermovement. The take-off phase began when CoM velocity became positive and continued until take-off, when the force dropped below a set threshold of 5 times SD of force produced during the flight phase, taken over 300 ms. The flight phase was defined as the time between the take-off and landing, during which the CoM velocity transitioned from positive to negative, reaching zero at the peak of the jump. Finally, the landing phase began when the negative CoM velocity returned to zero. The acceleration of the CoM was determined by dividing the net vGRF (absolute vGRF—body weight) by the participant’s body weight. The vertical CoM velocity profiles were obtained by integrating the acceleration of the CoM, which was calculated from the changes in velocity over time. 

#### 2.3.2. Markerless Data (MMPose)

OpenMMLab Pose Estimation Toolbox (MMPose; Version 1.3.2) was incorporated for markerless pose estimation purposes, utilizing a deep learning framework built on PyTorch. Participants’ body height was used to calibrate the markerless model, with the X–Z plane dividing the left and right sides of the body. The RTMDet model [24] was used for detecting the subject in each video frame [25]. Importantly, this model employs a gender-neutral design, ensuring that variability in measurements is not influenced by gender, thereby enhancing the reliability of comparisons between the markerless system and force plates. To ensure continuous subject tracking across the entire video sequence, especially during rapid movements or partial occlusions, a pixel-level object tracker (MOSSE from OpenCV) was combined with deepocsort for identity tracking. Subsequently, pose key-point estimation was performed using the Real-Time Model for Pose Estimation (RTMPose; [25]), which identified 14 anatomical landmarks and transformed them into 2D coordinates (i.e., X and Z). Figure 2 represents the localization of the anatomical key points on the subject. These key points were further refined using an Exponential Moving Average filter to reduce noise and exported as .tsv files for further analysis.

The CoM was determined using the same methodology as described by Aleksic et al. (2024) [14], based on the Dempster model (1955) for calculating segmental and overall CoM (i.e., full-body CoM). This approach applies specific segment weights to different body parts: foot (1.5%), lower leg (4.65%), upper leg (10%), and upper body (43%). Initially, the CoM for each segment (foot, lower leg, upper leg, and upper body) was calculated using X and Z coordinates and then combined to calculate the overall CoM. Subsequently, the CoM vertical velocity was calculated as the first derivative of the CoM position over time, using the following equation: $Vz\left(t\right)\approx \;\frac{Z\left(t+∆t\right)-Z(t-∆t)}{2∆t}$, where Z(*t*) represents the position vector of the CoM over time using Z_CoM_ coordinates. 

### 2.4. Variables

Figure 3 highlights eight key points corresponding to the key phases of the CMJ, based on the CoM vertical velocity (Vz-t), displacement (Z-t), and acceleration–time (az-t) traces, as captured by both the force plates and the markerless MMPose-based system. Subsequently, key temporal variables of the CMJ (Table 1) were defined and identified based on these eight specific key points. 

For the force plates system, point **a** indicates the point when vGRF drops below a set threshold ($\bar{X}$− 5*SD). Point **c** represents the minimum CoM Vz-t calculated by integration, before the point of take-off. Point **d** is the minimum CoM Z-t calculated by integration, before the point of take-off. Point **e** represents the maximum CoM Vz-t calculated by integration. Point **f** is the first point where F-t is in range (−10, 10)N. Point g indicates the maximum CoM Z-t calculated by integration. Point **h** is the last point where F-t is in the range (−10, 10)N. Point **i** is the point after landing where CoM Vz-t exceeds a value of zero. 

Similar key points were used for the MMPose system; however, they were also defined with reference to the CoM vertical displacement, and toe marker vertical displacement for the flight phase. Specifically, point **a** marks the point when the CoM vertical velocity (Vz-t) drops below 5% of its maximum, indicating the beginning of the unweighting phase. Point **c** represents the minimum vertical velocity before take-off, a key indicator of the end of the eccentric phase. Point **d** is the minimum Z-t value of the CoM before take-off, indicating the lowest position reached before the propulsion phase. Point **e** is the peak CoM vertical velocity. Points **f** and **h** identify the start and end of the flight phase, taken from the toe marker vertical displacement trajectory, by locating where toe height is less than 10% of its maximum on either side of the jump apex (point **g**). After landing, point **i** marks the minimum CoM Z-t. 

Figure 4 illustrates a correction applied to the calculation of jump height. Specifically, the jump height was adjusted for the height of the ankle at the moment of take-off. 

### 2.5. Statistical Analysis

The velocity–time traces obtained from the markerless MMPose-based system and force plates were compared to assess agreement between the two methods. Bland–Altman bias and limits of agreement (LoA) were calculated [26], along with root mean square error (RMSE) and Pearson’s correlation coefficient (r) to quantify the correlation between the methods. Correlation thresholds for r were defined in the following way [27]: r < 0.3 (negligible), r = 0.3–0.5 (low), 0.5–0.7 (moderate), 0.7–0.9 (high), and 0.9–1.0 (very high correlation). Bias, LoA, RMSE, and r were calculated using the MATLAB toolbox [28].

For the key temporal CMJ variables (phase duration, including the eccentric, propulsive, take-off, flight, and landing phases; maximum jump height; max CoM vertical velocity; and take-off CoM vertical velocity), the consistency between the two methods was further assessed using the intraclass correlation coefficient (ICC 3,1). Systematic differences between MMPose-derived and force plate-derived temporal measurements were evaluated using paired samples *t*-tests. In addition, Cohen’s d effect size was applied to quantify the practical significance of differences, with thresholds as follows [29]: d < 0.2 (trivial or no effect), d = 0.2–0.5 (small), d = 0.5–0.8 (moderate), d = 0.8–1.3 (large), and d > 1.3 (very large). Cohen’s d effect size and 95% confidence intervals (CI) were computed using the harmonic mean of the compared conditions’ standard deviations (SDs). An effect size of d = 0.20 was considered the minimum for practical importance. When the 95% CI included both positive and negative values, the effect was considered unclear; otherwise, the effect was deemed clear. Statistical significance was set at *p* < 0.05. Magnitude-based comparisons were conducted using a custom Excel spreadsheet (downloaded from https://www.sportsci.org/jour/03/wghtrials.htm, accessed on 18 September 2023), and other statistical analyses were performed using SPSS software (IBM SPSS Version 20.0, Chicago, IL, USA).

## 3. Results

Table 2 and Figure 5 present the comparison of the velocity–time traces for the center of mass (CoM) derived from the markerless MMPose-based system and the force plate. The analysis yielded a mean bias of 0.005 m/s (95% CI: −0.071 to 0.293), indicating a minimal average difference between the two systems. The limits of agreement (LoA) ranged from −0.253 m/s (95% CI: −0.514 to −0.151) to 0.263 m/s (95% CI: 0.128 to 1.099), showing the expected range within which differences in velocity measurements between the two methods may fall. The root mean square error (RMSE) was found to be 0.138 m/s (95% CI: 0.081 to 0.505), demonstrating a relatively low average error in velocity measurements. Furthermore, Pearson’s correlation coefficient (r) between the two systems was extremely high, at 0.992 (95% CI: 0.947 to 0.998), indicating a very strong linear relationship between the velocity–time traces obtained from the MMPose and force plate systems.

The descriptive statistics and validity metrics for temporal variables obtained from the markerless MMPose system and force plate are presented in Table 3. The results show strong agreement between the markerless MMPose system and the force plate across most CMJ performance variables, as indicated by high ICC values. The variables with the highest ICC values, reflecting excellent reliability, include jump height from flight time (ICC = 0.985, 95% CI: 0.974 to 0.992), propulsive phase (ICC = 0.974, 95% CI: 0.953 to 0.985), take-off phase (ICC = 0.971, 95% CI: 0.948 to 0.983), max CoM vertical velocity (ICC = 0.963, 95% CI: 0.934 to 0.979), and landing phase (ICC = 0.950, 95% CI: 0.912 to 0.972).

Significant systematic differences were observed between the two systems for several variables. The largest mean differences were found in the eccentric phase (−0.090 s, 95% CI: −0.099 to −0.082) and the take-off phase (−0.084 s, 95% CI: −0.092 to −0.077). Jump height also showed a notable mean difference of 0.068 m (95% CI: 0.054 to 0.082). Additional variables with significant but smaller mean differences included the propulsive landing phase (0.052 s, 95% CI: 0.013 to 0.091) and the landing phase (0.040 s, 95% CI: 0.015 to 0.066). 

Cohen’s d values, including 95% CI, are presented in Figure 6, showing the effect size magnitudes that are generally small to negligible for most dependent variables of interest. This indicates that while systematic differences exist, they are minor, with Cohen’s d values mostly within the range of −0.5 to 0.5.

## 4. Discussion

This study aimed to assess the level of agreement between a markerless MMPose-based system and a force plate system in measuring key performance indicators of the CMJ, such as vertical velocity, jump height, and jump phase durations. The main findings revealed a high level of agreement between the two systems, supporting the validity of the markerless approach in capturing detailed kinematic data during vertical jumps.

The results of the comparison of the velocity–time traces for the CoM derived from the markerless MMPose-based system and the force plate indicate a high level of agreement (r = 0.992) between the two systems in measuring the CoM velocity during the CMJ. The minimal bias (0.005 m/s) suggests negligible systematic differences between the methods, indicating that, on average, the markerless system closely aligns with the force plate data. Additionally, the narrow LoA ranging from −0.253 m/s to 0.263 m/s implies that individual measurements from the markerless system tend to be within a consistent range of the force plate measurements. This close agreement in individual values is important for ensuring that the markerless system’s outputs are both accurate and practically applicable. The low RMSE further supports the markerless system’s accuracy in tracking velocity, providing confidence that the velocity–time profile generated from the MMPose system closely resembles that derived from the force plate. This close match in velocity profiles supports the feasibility of using the MMPose markerless system as a reliable alternative for velocity-based analyses in sports performance and rehabilitation settings, where force plates may not always be accessible. 

Balsalobre-Fernández et al. (2023) reported that markerless or app-based measurements tend to overestimate jump height compared to traditional force plate systems, likely due to discrepancies in the detection of take-off and landing events [30]. The MMPose-based approach, however, addresses these discrepancies by deriving CoM velocity in a manner directly comparable to force plate outputs, minimizing such event-detection errors and enhancing reliability. Furthermore, the results for temporal variables also exhibit particularly high ICC values, such as jump height from flight time (ICC = 0.985), propulsive phase duration (ICC = 0.974), take-off phase duration (ICC = 0.971), maximum CoM vertical velocity (ICC = 0.963), and landing phase duration (ICC = 0.950). These results further confirm the markerless system’s consistency in capturing the duration of key CMJ phases, supporting its potential as a reliable alternative for tracking temporal performance metrics in settings where force plates may be less accessible. Such results align with findings from recent research supporting the reliability and validity of markerless systems for kinematic measurements in sports contexts [14,30].

Despite the solid inter-device agreement, small systematic differences between the two systems were noted in some temporal variables. The largest mean differences were observed in the eccentric phase (−0.090 s) and take-off phase (−0.084 s), where the markerless system consistently underestimated these values compared to force plates. Smaller yet significant mean differences appeared in the propulsive landing phase (0.052 s) and the landing phase (0.040 s). These discrepancies are likely related to methodological differences in detecting phase transitions, with markerless systems relying on computer vision algorithms that may slightly delay or advance the identification of start and end points of specific phases compared to ground reaction force detection. Previous studies have identified similar challenges in temporal measurement accuracy between methods, particularly in high-speed movements where computer vision systems may face limitations in real-time frame capture and processing [10,31]. The observed differences in this study, though systematic, are relatively minor in practical terms, as highlighted by effect size magnitudes, which mostly fall within the range of −0.5 to 0.5. While the markerless system’s measurements are not identical to those of the force plate, this range suggests that the effect sizes are small to negligible, implying that the two systems are functionally comparable for most CMJ performance variables. 

A notable mean difference was only found for jump height (0.068 m) as measured by vertical CoM displacement, where the markerless system consistently overestimated jump height compared to the force plate. This is likely due to differences in the jump height calculation methods, since the markerless system directly tracks the CoM trajectory, whereas the force plate method relies on the double integration of force data. This process can introduce cumulative drift errors, especially in movements with rapid fluctuations [31,32]. This limitation of force plate-based calculations is well documented, with research suggesting that numerical integration often requires additional corrections, such as filtering or baseline adjustments, to reduce error in displacement estimates [32]. When the markerless system’s jump height was adjusted for ankle height at take-off and compared with the force plate height obtained via the impulse–momentum approach, the difference between systems was negligible (0.005 m). This aligns with findings from Bridgeman et al. (2024), who demonstrated that minor adjustments to correct for pre-take-off height discrepancies could enhance the comparability of jump height measurements across different systems [33]. Additionally, the jump height from flight time variable showed high agreement between the two systems (ICC = 0.985), with negligible mean difference. Flight time-based measurement is a commonly used field method, and previous studies, such as those by Wade et al. (2020), have found it reliable for approximating jump height when direct measurement of CoM displacement is unavailable [3]. However, the flight time approach may slightly underestimate jump height due to unaccounted for heel-lift at take-off, a consideration relevant for both systems and supported by recent studies exploring field-based measurement enhancements [3].

The observed agreement between MMPose and force plate measures, along with minor discrepancies that are generally small in magnitude, support the markerless system’s potential as a valid alternative for CMJ analysis. While differences in specific variables may be statistically significant, they are of limited practical relevance given the small effect sizes observed. This finding supports the conclusions of previous studies [14,19] that demonstrated that markerless systems could provide high-quality, reliable data in field settings without the logistical and financial burdens of force plates. These advantages are particularly beneficial in high-volume or time-constrained environments, such as applied team sports or rehabilitation settings, where ease of use and quick setup are crucial. Compared to simpler jump analysis tools such as the My Jump® app, which primarily estimates jump height based on flight time duration [18], the MMPose-based system offers a more comprehensive analysis of CMJ performance. Rather than focusing solely on peak metrics, this markerless approach allows for a detailed overview of the CMJ phase durations and enables velocity-based analysis across all time points. The ability to monitor jump kinematics in this detailed manner allows coaches and practitioners to assess not only the overall jump performance (e.g., jump height and take-off velocity) but also specific technique-related variables (e.g., joint angle changes, eccentric duration, or peak propulsive velocity). This level of detail is particularly valuable for individualized training adjustments and rehabilitation monitoring, as different phases of the CMJ may be emphasized based on an athlete physical readiness or recovery status [34,35]. For example, research indicates that athletes who have met traditional return-to-sport discharge criteria after anterior cruciate ligament (ACL) reconstruction may still display significant asymmetries in the propulsive phase or joint angle patterns [21,36]. These insights could be instrumental in reducing injury risk and enhancing overall athletic performance.

While the findings of this study support the use of the markerless MMPose-based system as a reliable alternative to force plates, particularly in field-based settings, it is important to acknowledge certain study limitations. One limitation was the choice to measure CoM displacement using the direct velocity–time integration rather than through the double integration of the CoM acceleration derived from the force plate. This decision was made to maintain the integrity of the velocity signals for comparison. While double integration is commonly used for calculating displacement from force plate data, it can introduce noise and drift, potentially distorting the original signal [32]. By focusing on the first derivative (i.e., velocity) as derived directly from both systems, we tried to mitigate potential discrepancies that would emerge from a second integration step. Several studies emphasize the challenge of signal distortion in double integration for MoCap data [32,37]; therefore, by limiting our analysis to the velocity domain, we wanted to avoid introducing systematic or signal errors into our comparison. Future studies should be focused on methods to reduce drift in CoM displacement calculation from force plates, perhaps by applying correction techniques. Such approach could allow for an even broader comparison of markerless and force plate data, covering both velocity and displacement domains without introducing integration errors.

Additionally, given that the accuracy of the markerless MMPose framework was assessed in the context of a CMJ task, which is a relatively straightforward movement, future studies should focus on exploring more complex, multi-directional, or rotational movement patterns as well. Furthermore, this study’s limited sample size may restrict the generalizability of the findings to different populations (e.g., injured athletes or athletes of diverse athletic backgrounds or training levels); therefore, future studies should address the feasibility of MMPose across a broader range of populations. It is also essential to explore how MMPose performs in real-world scenarios, where environmental variables like lighting and background complexity may pose challenges in the accuracy of pose detection.

## 5. Conclusions

In conclusion, the findings of the present study suggest that the MMPose-based markerless motion capture system offers a highly accurate alternative to traditional force plates for measuring key performance indicators of the CMJ. Its strong agreement with force plate-derived measurements validates its use for velocity profiling, and its potential to capture phase-specific and continuous movement data provides an edge over simpler markerless smartphone-based jump analysis solutions. This advancement in markerless technology could pave the way for more widespread kinematic assessments in field-based settings, promoting data-driven approaches to training and recovery where force plates are less accessible. This study adds to the growing body of evidence validating markerless motion capture for vertical jump assessment and reinforces its suitability for both field and laboratory settings [14,31,33].

As markerless technology continues to advance, its practicality and user-friendliness are expected to improve, which may open doors to more in-depth analyses of other common tasks that are used in applied sports and rehabilitation settings. These could include more complex actions like unilateral jumps, horizontal jumps, and other types of dynamic movements. By expanding its utility across diverse assessment modalities, markerless motion capture could provide athletes, coaches, and clinicians with a comprehensive toolkit to support optimal performance and recovery strategies. Thus, the continued evolution of markerless systems holds considerable promise for enhancing movement analysis and broadening accessibility to data-driven methodologies in real-world settings.

## Figures and Tables

**Figure 1 life-14-01641-f001:**
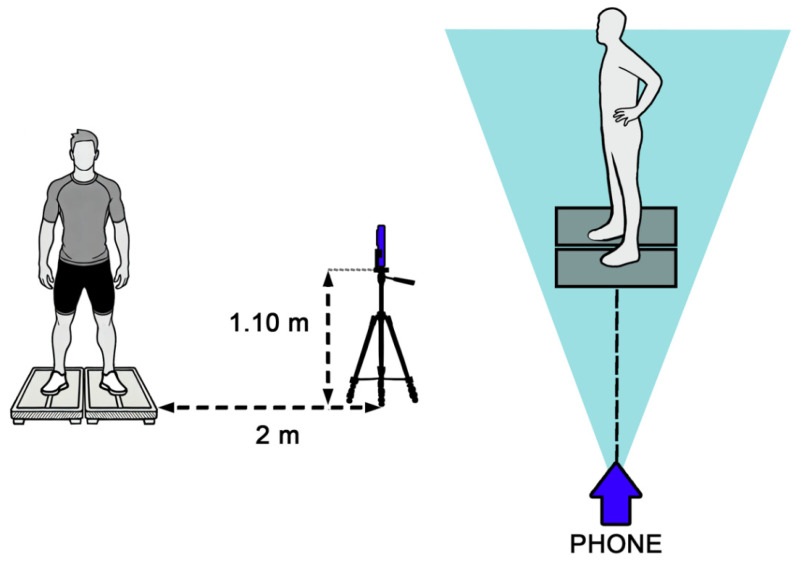
Illustration of the experimental set-up.

**Figure 2 life-14-01641-f002:**
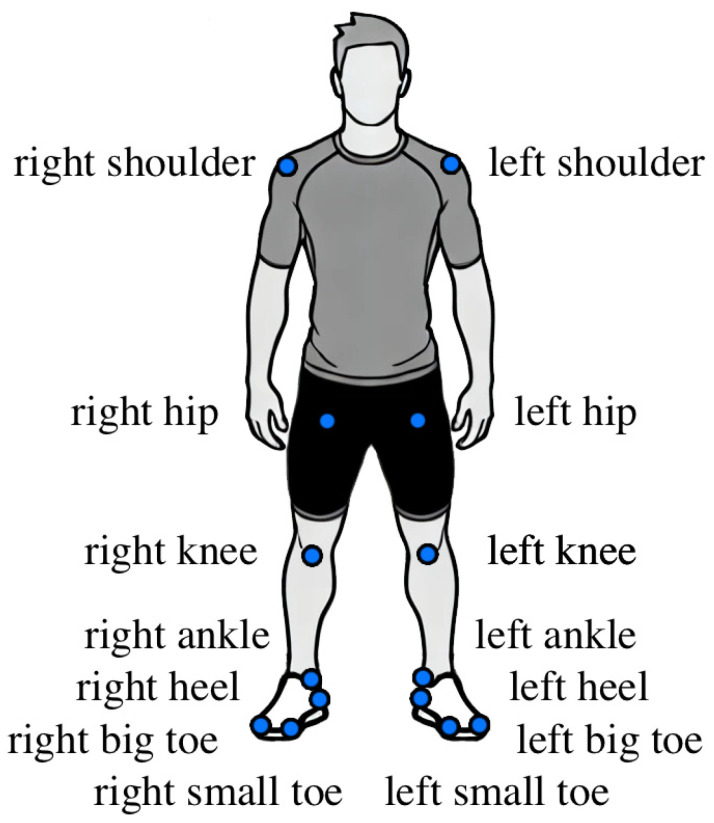
Localization of anatomical landmarks on the subject in MMPose.

**Figure 3 life-14-01641-f003:**
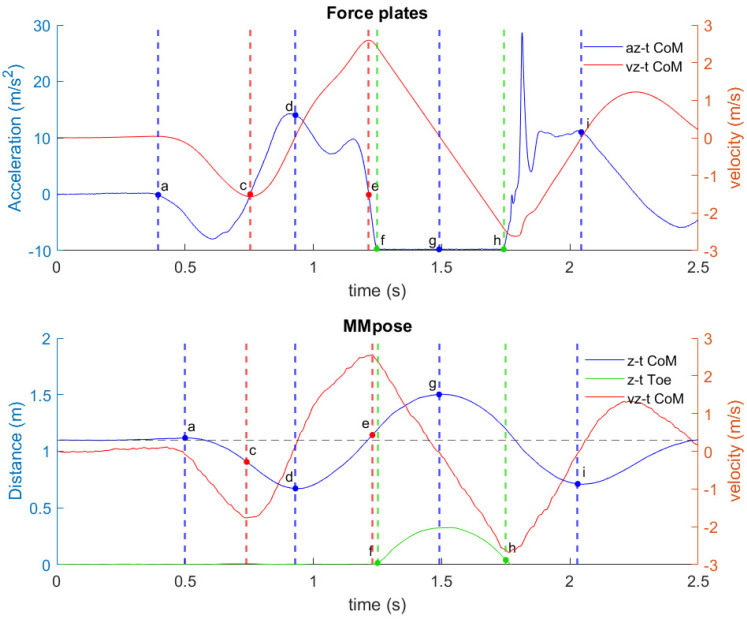
Key points of the CMJ signal based on the Vz-t (vertical velocity–time), az-t (vertical acceleration–time), and z-t (vertical displacement–time) profiles for force plates and MMPose. Blue line—CoM acceleration signal (force plates) and CoM displacement signal (markerless system), red line—CoM vertical velocity signal, green line—toe vertical displacement signal.

**Figure 4 life-14-01641-f004:**
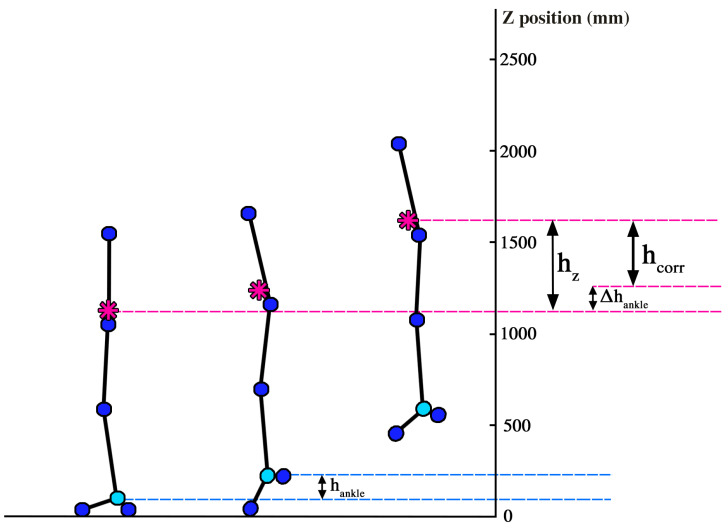
Jump height correction based on ankle height at take-off. Pink symbol—center of mass (CoM); pink dashed line—CoM height; blue dashed line—ankle marker height; h_z_—uncorrected jump height, measured based on the CoM vertical displacement relative to the initial resting position; h_ankle_—ankle height at take-off relative to the initial resting position; h_corr_—jump height adjusted to the ankle height at take-off.

**Figure 5 life-14-01641-f005:**
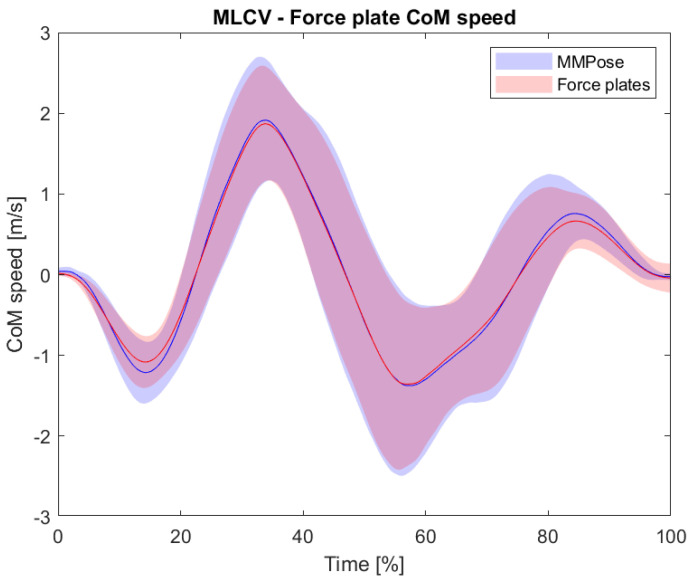
Illustrative comparison of the velocity–time traces for the center of mass (CoM) derived from the markerless MMPose-based system and the force plates. The solid blue lines represent data from the markerless system, and the red lines represent data from the force plate system. The shaded areas represent variability (95% confidence intervals).

**Figure 6 life-14-01641-f006:**
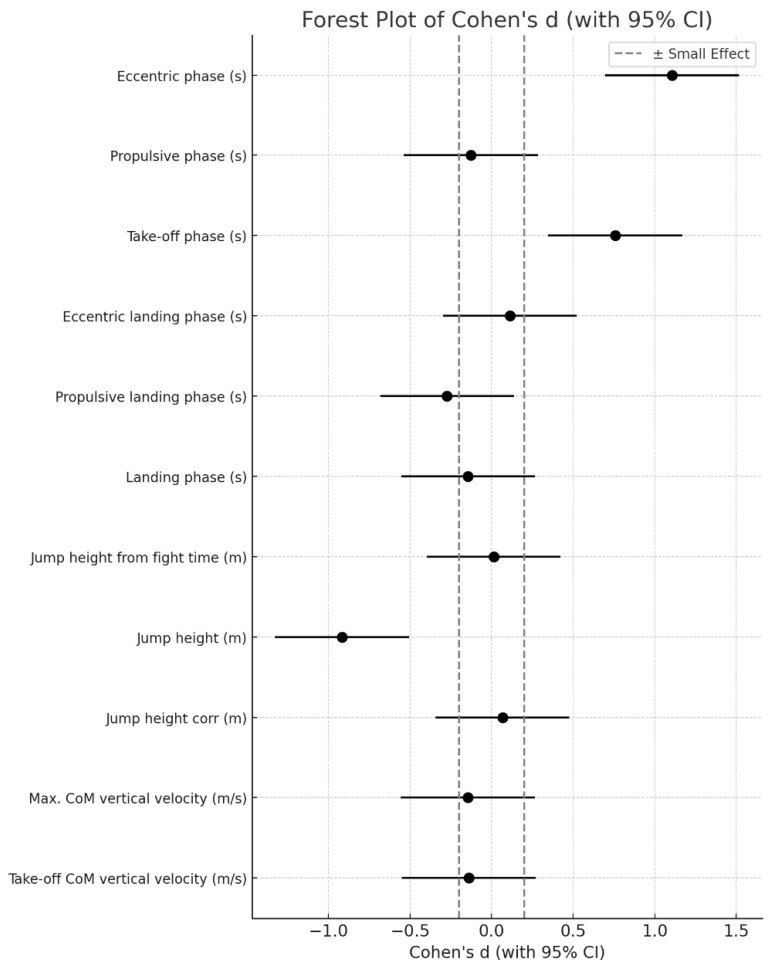
Cohen’s d values (with 95% confidence intervals) for CMJ temporal variables.

**Table 1 life-14-01641-t001:** Definition of key CMJ temporal variables that were analyzed in this study.

Variable	Explanation
Eccentric phase (s)	The time interval between point **a** (start of the movement) and point **d** (end of the eccentric phase) in the CoM Z-t trajectory.
Propulsive phase (s)	The time interval between point **d** (end of the eccentric phase) and point **f** (take-off) in the CoM Z-t trajectory.
Take-off phase (s)	The time interval between point **a** (start of the movement) and point **f** (take-off) in the CoM Z-t trajectory.
Landing Eccentric phase (s)	The time interval between point **h** (landing) and point **i** (end of landing eccentric phase) in the CoM Z-t trajectory.
Landing Concentric phase (s)	The time interval between point **i** (end of landing eccentric phase) and point **j** (end of landing concentric phase) in the CoM Z-t trajectory.
Landing phase (s)	The time interval between point **h** (landing) and point **j ** (end of landing concentric phase) in the CoM Z-t trajectory.
Jump height from flight time (m)	Jump height calculated based on the flight time.
Jump height (m)	Maximum vertical displacement of the CoM relative to the initial resting position.
Jump height corr (m)	Jump height adjusted for ankle height at take-off.
Max CoM vertical velocity (m/s)	Maximum vertical velocity of the CoM during the movement.
Take-off CoM vertical velocity (m/s)	Vertical velocity of the CoM at the moment of take-off.

**Table 2 life-14-01641-t002:** Comparison between force plate-derived vertical velocity time trace (m/s) and those obtained from markerless solution: Bias, limits of agreement, RMSE, and correlation coefficients (mean (min ÷ max)).

Bias	LoA Lower	LoA Upper	RMSE	Correlation
0.005 (−0.071 ÷ 0.293)	−0.253 (−0.514 ÷ −0.151)	0.263 (0.128 ÷ 1.099)	0.138 (0.081 ÷ 0.505)	0.992 (0.947 ÷ 0.998)

Note: LoA—limits of agreement; RMSE—root mean square error.

**Table 3 life-14-01641-t003:** Descriptive statistics (mean (SD)), mean difference (95%CI), t-value, and ICC (95%CI) for temporal variables obtained by the markerless system and the force plates.

	Markerless	Force Plate				
Variable	Mean (SD)	Mean (SD)	Mean Difference (95% CI)	t(47)	*p*	ICC (95% CI)
Eccentric phase (s)	0.392 (0.073)	0.482 (0.089)	−0.090 (−0.099 ÷ −0.082)	−21.67 **	<0.001	0.937 (0.891 ÷ 0.964)
Propulsive phase (s)	0.275 (0.039)	0.27 (0.039)	0.006 (0.003 ÷ 0.008)	4.47 **	<0.001	0.974 (0.953 ÷ 0.985)
Take-off phase (s)	0.668 (0.103)	0.752 (0.118)	−0.084 (−0.092 ÷ −0.077)	−21.73 **	<0.001	0.971 (0.948 ÷ 0.983)
Landing eccentric phase (s)	0.291 (0.1)	0.303 (0.115)	−0.012 (−0.027 ÷ 0.003)	−1.58	0.122	0.882 (0.799 ÷ 0.932)
Landing concentric phase (s)	0.486 (0.193)	0.433 (0.195)	0.052 (0.013 ÷ 0.091)	2.70 **	0.010	0.762 (0.612 ÷ 0.860)
Landing phase (s)	0.776 (0.284)	0.736 (0.271)	0.040 (0.015 ÷ 0.066)	3.17 **	0.003	0.950 (0.912 ÷ 0.972)
Jump height from flight time (m)	0.285 (0.077)	0.286 (0.083)	−0.001 (−0.005 ÷ 0.003)	−0.26	0.799	0.985 (0.974 ÷ 0.992)
Jump height (m)	0.363 (0.079)	0.295 (0.069)	0.068 (0.054 ÷ 0.082)	9.72 **	<0.001	0.787 (0.650 ÷ 0.875)
Jump height corr (m)	0.27 (0.076)	0.275 (0.074)	−0.005 (−0.016 ÷ 0.007)	−0.85	0.400	0.864 (0.770 ÷ 0.921)
Max CoM vertical velocity (m/s)	2.502 (0.309)	2.457 (0.307)	0.045 (0.020 ÷ 0.069)	3.67 **	0.006	0.963 (0.934 ÷ 0.979)
Take-off CoM vertical velocity (m/s)	2.346 (0.339)	2.299 (0.339)	0.047 (0.014 ÷ 0.080)	2.86 **	0.400	0.944 (0.902 ÷ 0.968)

Note: ** *p* < 0.01.

## Data Availability

Dataset available on request from the authors. The raw data supporting the conclusions of this article will be made available by the authors on request.

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
