# Peer review of "Advancing Field-Based Vertical Jump Analysis: Markerless Pose Estimation vs. Force Plates"

_life, 2024, doi:10.3390/life14121641_

Round 1

Reviewer 1 Report

Comments and Suggestions for Authors

Review opinion in attachment

Author Response

Comment 1: The authors of the study decided to compare the reliability of vertical jump measurements from a dynamometric platform and using MMPose. They argue that using a dynamometric plate requires more organization and costs than MMPose., because MMpose is an open source toolkit based on Pytorch (open software but not free) for estimating body position. It contains a rich set of algorithms for estimating the human body position in 2D at 133 key points positioning the entire human body.
In the introduction, the authors convinced the reader about the role of knowledge of lower body power, especially for the needs of sports and rehabilitation. They argue that advances in the field of artificial intelligence have revolutionized the science of human movement, offering cost-effective and easily accessible alternatives to markerless assessment of human body position, which can successfully replace traditional expensive and calibration-requiring measurement systems based on 3D markers. For this purpose, they used numerous examples from the literature on the subject of the research, informing that parkless systems, especially MMPose, are much more accessible and therefore can be more widely used.
After justifying the role of MMpose, the authors set themselves the goal of examining the differences between the markerless system and the dynamometric plate in measuring key time and kinematic variables, such as vertical velocity, jump height and vertical jump velocity. In order to learn about the differences, the authors conducted the correct research, taking into account the selection of recreational athletes in accordance with the guidelines of the Institutional Ethical Review Board of the University (approval no. 02-848/23-2; date: May 5, 2023) and the Declaration of Helsinki, the necessary procedures taking into account the measurement instruments and methodological procedure. Therefore, the procedure for selecting subjects, the measurement session does not raise any objections. Similarly, the processing of data from the dynamometric plate and MMPose was assessed without reservations.
Appropriate statistical methods were selected to investigate the significance of differences in vertical jump results using the dynamometer plate and MMPose, based on which the Authors concluded that the results indicate a high agreement between the markerless MMPose system and the dynamometer plate for most of the vertical jump variables.
In the discussion, the Authors of the study confirmed the use of the markerless system based on MMPose as a reliable alternative to dynamometer plates, which can be used especially in field conditions, with less financial and organizational effort, and confirm these results in the conclusions. They argue that the use of MMPose can have a wider application than dynamometer plates covering more complex activities such as unilateral jumps, horizontal jumps and other types of dynamic human movements. By extending its utility to different assessment methods, markerless motion capture can provide athletes, coaches and clinicians with a comprehensive set of tools to support optimal strategies for assessing and shaping power and recovery. The above work deserves attention for the applied literature on the subject, almost all of which was published in the 21st century, and a large part of it over the last few years. Overall, the work is correct, and its results can convince many athletes, trainers, and rehabilitators to use MMPose in practice in their professions and everyday professional practice.

Response 1: Thank you for your thoughtful and positive feedback on our manuscript. We are very pleased that you found our study to be well-conducted and that its results hold potential for practical application in sports and rehabilitation settings. We appreciate your recognition of the importance of our work in comparing the markerless MMPose system to traditional force plates and highlighting the potential of MMPose for broader, field-based applications. Your acknowledgment of our methodological rigor is greatly valued. Your comments reinforce our belief in the significance of providing cost-effective and accessible alternatives like MMPose to enhance movement analysis in sports science and rehabilitation. Should you have additional suggestions or insights on further enhancing the practical implications or addressing specific audiences, we would be happy to consider them for inclusion in the manuscript. Thank you once again for your encouraging remarks and constructive evaluation of our work!

Reviewer 2 Report

Comments and Suggestions for Authors

Thank you to the authors for their research.

However, I have some comments:

I suggest optimizing the summary. I think it is not appropriate to detail the methodology in the summary; it is enough to name it. It is not necessary to provide essential data; it is enough to provide the main thesis. The summary should arouse intrigue but not reveal the entire content.

The introduction should present an argument for the relevance of this research and only then formulate the goal. It is necessary to answer the questions: What has already been done, what has not yet been done, what needs to be researched, and who needs this research?

The differences between men and women in the high jump event can be explained through physiological, biomechanical, and anatomical factors that influence performance. High jump performance is a result of the intricate balance between power, technique, and body control for both sexes. Therefore, the study group should be characterized by homogeneity (normal distribution) in the evaluated indicators. I recommend analyzing the groups of men and women separately and performing an assessment of the distribution of their data (e.g., the Kolmogorov-Smirnov test).

The methods used by the author to assess the difference in data do not allow for making reasonable assumptions about their reliable difference or similarity. I suggest choosing appropriate tools (e.g., Student's t-test).

I think it is necessary to present the limitations of the study.

Sincerely.

Author Response

Comments 1: I suggest optimizing the summary. I think it is not appropriate to detail the methodology in the summary; it is enough to name it. It is not necessary to provide essential data; it is enough to provide the main thesis. The summary should arouse intrigue but not reveal the entire content.
The introduction should present an argument for the relevance of this research and only then formulate the goal. It is necessary to answer the questions: What has already been done, what has not yet been done, what needs to be researched, and who needs this research?
The differences between men and women in the high jump event can be explained through physiological, biomechanical, and anatomical factors that influence performance. High jump performance is a result of the intricate balance between power, technique, and body control for both sexes. Therefore, the study group should be characterized by homogeneity (normal distribution) in the evaluated indicators. I recommend analyzing the groups of men and women separately and performing an assessment of the distribution of their data (e.g., the Kolmogorov-Smirnov test).
The methods used by the author to assess the difference in data do not allow for making reasonable assumptions about their reliable difference or similarity. I suggest choosing appropriate tools (e.g., Student's t-test). I think it is necessary to present the limitations of the study.

Response 1: We greatly appreciate your thoughtful feedback and have addressed the points raised as follows: 
- To optimize the summary, we revised it to provide a concise overview of the study, focusing on the key findings and significance while omitting detailed methodological descriptions as you suggested. The updated summary highlights the comparison between markerless motion capture and force plates and briefly underscores the results of our findings (page 1, paragraph 14-32).
- Regarding the Introduction, after carefully reviewing this section, we confirm that the text emphasizes the relevance of the study by detailing advancements in markerless motion capture technology and its limitations compared to traditional systems. Specifically, the text outlines what has been achieved (page 2, paragraph 64-88), identifies the gap in research (page 2, paragraph 90-109), and explains the necessity of this study to improve accessibility and scalability in performance assessments (page 3, paragraph 110-122). Additionally, the text also highlights the practical benefits for researchers, clinicians, and practitioners in sports science and rehabilitation (page 3, paragraphs 116-119, and 127-129).    
- In terms of participant demographics, we recognize the importance of comprehensive data for generalizability. Therefore, we have provided additional details in the Methods section, specifying the number of male and female participants along with mean and standard deviation values for age, height, and body mass for each gender (page 3, lines 132-136). However, as the primary goal of this study is to compare the accuracy of two measurement systems (i.e., markerless motion capture and force plates), we have maintained this focus in the Results section. While participant performance could certainly vary by gender, our analysis is focused on the systems' accuracy rather than gender-specific performance outcomes. Notably, the markerless model used in this study is designed to be gender-neutral, further supporting our approach. This ensures that our conclusions emphasize the reliability and applicability of the systems itself, independent of gender. For better clarity, we have now included this explanation in the Methods section (page 5, lines 195-198).
- We also appreciate your suggestion regarding statistical tools. Paired t-tests were employed to evaluate systematic differences between the two systems for key temporal variables, as detailed in the Methods section (page 8, line 272).
- Finally, thank you for highlighting the importance of addressing study limitations. We completely agree with your suggestion and have expanded the Discussion section to include a more detailed account of the study’s limitations (page 13, lines 447-455). These limitations pertain to the nature of the CMJ task, limited sample size and the need to conduct more research outside laboratory environments to explore how MMPose performs in real-world scenarios, where variables such as lighting or background complexity could impact the accuracy of pose detection.
- Additionally, we noticed that the order of tables was mixed during the reformatting of the manuscript in the Life template. Table 3 has now been relocated to its correct section in the manuscript (page 10, lines 313–315).
We hope these edits to the original manuscript address your concerns and enhance the overall quality of the paper.

Reviewer 3 Report

Comments and Suggestions for Authors

The reviewed article concerns the comparison of a modern markerless system for vertical jump analysis with traditional force plates. The study was conducted on a group of 12 healthy participants, demonstrating that MMPose is a cost-effective alternative for movement analysis in field conditions, offering high accuracy in assessing vertical velocity, jump height, and key jump phases. The use of markerless motion capture supported by artificial intelligence algorithms is an innovative approach with the potential to reduce the costs of biomechanical analysis. The study was conducted under well-controlled laboratory conditions in line with the steps of the scientific method, and the results were presented transparently. A strong correlation (r=0.992) between the markerless system and force plates suggests high validity of the method. Practical possibilities for using the results in sports analysis and rehabilitation in environments with limited access to advanced equipment were highlighted.

Remarks: The sample size is insufficient for generalizing the results. Inclusion criteria poorly visible. Expanding the study group to include athletes from various disciplines and training levels, as well as individuals with lower limb injuries, would enhance reliability and representativeness. It would be beneficial to provide information on the training level of participants. Jump height results show a tendency for overestimation by MMPose. There is no explanation of whether such differences might affect the practical application in sports analysis. The article focuses exclusively on CMJ. Analyzing more complex movements, such as unilateral or horizontal jumps, could confirm the system's versatility. It would be worthwhile to extend the research to other biomechanical movements to better understand the system's potential applications. Further work on validating and optimizing the system could make markerless fully comparable to traditional methods across a wider range of applications. Add practical implications and limitations.

Author Response

Comments 1: The sample size is insufficient for generalizing the results. Inclusion criteria poorly visible. Expanding the study group to include athletes from various disciplines and training levels, as well as individuals with lower limb injuries, would enhance reliability and representativeness. It would be beneficial to provide information on the training level of participants. Jump height results show a tendency for overestimation by MMPose. There is no explanation of whether such differences might affect the practical application in sports analysis. The article focuses exclusively on CMJ. Analyzing more complex movements, such as unilateral or horizontal jumps, could confirm the system's versatility. It would be worthwhile to extend the research to other biomechanical movements to better understand the system's potential applications. Further work on validating and optimizing the system could make markerless fully comparable to traditional methods across a wider range of applications. Add practical implications and limitations.

Response 1: We appreciate your thorough feedback and have address your suggestions as follows:
- In terms of the study participants, the sample size for this study was determined using G*Power analysis, as stated in the Methods section (page 3, lines 143-145), to ensure sufficient statistical power for comparing the two measurement systems. We have clarified the inclusion criteria in the revised manuscript to improve visibility and transparency (page 3, lines 136-140). As the primary goal of this study was to compare the accuracy of the markerless MMPose system with force plates, our focus was on assessing the agreement between these systems rather than on participant performance outcomes. While we acknowledge that athletic background or training level may influence performance metrics, these factors were not of primary interest for this study. Instead, our analysis aimed to evaluate the reliability and practicality of the two systems across a standardized task.
However, we have revised the manuscript to address the limitations pertaining to the sample, highlighting that the small sample in this study may limit the generalizability of the findings to other populations, such as injured athletes or those with diverse athletic backgrounds and training levels. We also emphasized the importance of future studies involving a broader range of populations (page 13, lines 447-455).
- We acknowledge that MMPose demonstrated a slight tendency to overestimate jump height compared to force plates (page 12, lines 381-390). However, we also proposed a solution by adjusting the markerless system’s jump height for ankle height at take-off, which yielded a very small difference between the systems (0.005 m). We have included this in the Discussion session for further clarification (page 12, line 390-397).
- We have expanded the Limitations section to emphasize the need for future research exploring more complex movements, such as rotational or multi-directional movements, to assess the versatility of the markerless system. Additionally, we highlighted the importance of evaluating MMPose in real-world scenarios, where environmental variables like lighting and background complexity may pose challenges. We also noted that future studies should include more diverse populations, such as individuals with varying athletic backgrounds, training levels, and lower limb injuries, to enhance the generalizability and representativeness of findings (page 13, lines 447-455).
- Additionally, we noticed that the order of tables was mixed during the reformatting of the manuscript in the Life template. Table 3 has now been relocated to its correct section in the manuscript (page 10, lines 313–315).

Round 2

Reviewer 2 Report

Comments and Suggestions for Authors

I appreciate the writers' reactions to my notes and their modifications to the manuscript.

The updated manuscript, in my opinion, satisfies every condition. It should be published, in my opinion.

Warm regards.